# A Mathematical Framework for Quantifying Transferability in Multi-source Transfer Learning

**Xinyi Tong**
Tsinghua-Berkeley Shenzhen Institute
Tsinghua University
`txy18@mails.tsinghua.edu.cn`

**Xiangxiang Xu**[*]
Massachusetts Institute of Technology
`xuxx@mit.edu`

**Shao-Lun Huang**[†]
Tsinghua-Berkeley Shenzhen Institute
Tsinghua University
`shaolun.huang@sz.tsinghua.edu.cn`

**Lizhong Zheng**
Massachusetts Institute of Technology
`lizhong@mit.edu`

## Abstract

Current transfer learning algorithm designs mainly focus on the similarities between source and target tasks, while the impacts of the sample sizes of these tasks are often not sufficiently addressed. This paper proposes a mathematical framework for quantifying the transferability in multi-source transfer learning problems, with both the task similarities and the sample complexity of learning models taken into account. In particular, we consider the setup where the models learned from different tasks are linearly combined for learning the target task, and use the optimal combining coefficients to measure the transferability. Then, we demonstrate the analytical expression of this transferability measure, characterized by the sample sizes, model complexity, and the similarities between source and target tasks, which provides fundamental insights of the knowledge transferring mechanism and the guidance for algorithm designs. Furthermore, we apply our analyses for practical learning tasks, and establish a quantifiable transferability measure by exploiting a parameterized model. In addition, we develop an alternating iterative algorithm to implement our theoretical results for training deep neural networks in multi-source transfer learning tasks. Finally, experiments on image classification tasks show that our approach outperforms existing transfer learning algorithms in multi-source and few-shot scenarios.

## 1 Introduction

Transfer learning is nowadays an active research area in machine learning focusing on solving target learning tasks by the knowledge of learnable source tasks. The transferability between source and target tasks is the central topic in transfer learning for understanding the knowledge transferring mechanisms and the algorithm designs [1]. In general, the transferability can be affected by several factors, including: (i) the similarities between source tasks and the target task [2]; (ii) the sample sizes of the tasks; and (iii) the complexity or dimensionality of the machine learning model. Most of the existing transfer strategies are designed based on how similar the source and target tasks are [2, 3], without considering the impacts of the training sample sizes or the complexity of the models.

---

[*]Work performed at Tsinghua-Berkeley Shenzhen Institute

[†]Corresponding author

35th Conference on Neural Information Processing Systems (NeurIPS 2021).

In theoretical analyses [4, 5, 6], sample sizes and model complexity are often included in deriving upper bounds for the transferability or the performance of transfer learning algorithms. However, it is pointed out that such bounds derived under general learning settings are often relatively loose under numerical simulations [7], and hence the algorithms designed by directly applying theoretical results can hardly achieve satisfactory performance in practical applications. Thus, the gap between theory and practice opposes the fundamental understandings of transfer learning algorithms.

In this paper, we propose a mathematical framework to investigate the transferability in multi-source transfer learning problem, and establish a quantifiable transferability measure for practical learning tasks. Specifically, for given source tasks, we learn the target task by a class of learning model which linearly combines the models learned from individual tasks by some designable coefficients. In addition, the performance of this combined model is measured by the empirical risk of only the *testing data* of the target task, considered as the testing loss. Then, we adopt the optimal combining coefficients that achieve the minimum testing loss as the transferability measure, which illustrates the contribution of each model in learning the target task, and effectively quantifies the knowledge transferable among different tasks.

In our development, we establish an analytical solution of the transferability measure, which is jointly quantified by sample sizes, model complexity, and a similarity measure between source and target tasks. In particular, we demonstrate that the transferability of a particular source task is typically proportional to the number of samples and the measure of similarity to the target task, and is inversely proportional to the model complexity. This coincides with the intuition that when more training samples are available for a source task that is highly similar to the target task, more knowledge will be transferable from the source task to the target task. On the other hand, when the model is very complicated or high-dimensional, it is typically harder to train the model well, and less knowledge can be acquired and transferred. More importantly, our theoretical results can be applied for designing effective and efficient algorithms for real transfer learning problems, which are especially useful for multi-source transfer learning with a large number of source tasks that are generally difficult to deal with.

The contribution of this paper can be summarized as follows:

- We propose a mathematical framework for transfer learning analyses, and establish a transferability measure on discrete data, quantified by the number of samples, the complexity of the model, and the $\chi^2$-distance between source and target tasks.

- We extend the transferability analyses to the continuous data, and establish a similar transferability measure that can be evaluated in practical tasks, by exploiting paramerized models.

- We apply our theoretical results to develop an iterative algorithm for training deep neural networks in general supervised transfer learning scenarios. Moreover, our algorithm can be practically applied for multi-source transfer learning.

- The experiments in real datasets validate our proposed algorithm, in which we show that our approach outperforms many existing transfer learning algorithms.

Due to the space limitations, the proofs of theorems and propositions are presented in the supplemental material.

## 2 Problem Formulation and Analysis

Let $X$ and $Y$ be the random variables denoting the data and label with domains $\mathcal{X}$ and $\mathcal{Y}$, respectively, and let $\mathcal{P}$ denote the set of all distributions on $\mathcal{X} \times \mathcal{Y}$. For the convenience of illustration, here we assume $X$ to be discrete, and will extend our analyses to continuous cases later. Throughout our analyses, we will use $\mathcal{A}_k \triangleq \{(\alpha_0, \ldots, \alpha_k) \colon \sum_{i=0}^{k} \alpha_i = 1, \alpha_i \geq 0, i = 0, \ldots, k\}$ to denote the $k$-dimensional simplex.

### 2.1 Single-Source Transfer Learning

To begin, we consider the transfer learning setting with one source task and one target task, denoted as task 1 and 0, respectively. Specifically, for each task $i = 0, 1$, we assume that $n_i$ training

samples $\{(x_\ell^{(i)}, y_\ell^{(i)})\}_{\ell=1}^{n_i}$ are i.i.d. generated from some underlying joint distribution $P_{XY}^{(i)} \in \mathcal{P}$ with[3] $P_{XY}^{(i)}(x, y) > 0$, for all $x, y$, and the empirical distributions $\hat{P}_{XY}^{(i)} \in \mathcal{P}$ of the samples are defined as

$$\hat{P}_{XY}^{(i)}(x, y) \triangleq \frac{1}{n_i} \sum_{\ell=1}^{n_i} \mathbb{1}\{x_\ell^{(i)} = x, y_\ell^{(i)} = y\}, \tag{1}$$

where $\mathbb{1}\{\cdot\}$ denotes the indicator function [8]. Then, the empirical distributions $\hat{P}_{XY}^{(0)}$ and $\hat{P}_{XY}^{(1)}$ can be regarded as the models learned from the target task and the source task, respectively, when all the entries of the mass functions are required to determine.

To develop the transferability measure, our proposed framework focuses on a convex combination of both learned models[4]:

$$Q_{XY}^{(\alpha_0, \alpha_1)}(x, y) \triangleq \alpha_0 \hat{P}_{XY}^{(0)}(x, y) + \alpha_1 \hat{P}_{XY}^{(1)}(x, y), \quad \text{for all } (x, y) \in \mathcal{X} \times \mathcal{Y}, \tag{2}$$

where $(\alpha_0, \alpha_1) \in \mathcal{A}_1$ are parameters to be designed. Notice that these parameters characterize the knowledge transferred from the source task to target task, and the designing of these parameters will be affected by the sample sizes and the task similarities, which essentially leads to a transferability measure adjusted by the sample complexity.

Then, the performance of the model $Q_{XY}^{(\alpha_0, \alpha_1)}$ is evaluated by the testing loss, measured by its empirical risk on the testing data of target task. Conventionally, such empirical risk is often computed by the logarithm loss. However, the logarithm risk can be ill-defined[5] in our setting. Therefore, we alternatively apply the referenced $\chi^2$-distance as the measure, defined as follows.

**Definition 1.** *Given a reference distribution $R_{XY}$, for any distribution $P_{XY}$ and $Q_{XY}$, the referenced $\chi^2$-distance between them is defined as*

$$\chi_{R_{XY}}^2(P_{XY}, Q_{XY}) \triangleq \sum_{x \in \mathcal{X}, y \in \mathcal{Y}} \frac{(P_{XY}(x, y) - Q_{XY}(x, y))^2}{R_{XY}(x, y)}.$$

*Specifically, we denote $\chi^2(P_{XY}, Q_{XY}) \triangleq \chi_{P_{XY}}^2(P_{XY}, Q_{XY})$, which corresponds to the Pearson $\chi^2$-divergence.*

We choose the underlying target distribution $P_{XY}^{(0)}$ as the reference, and define the testing loss as the averaged Pearson $\chi^2$-divergence

$$L_{\text{test}}^{(\alpha_0, \alpha_1)} \triangleq \mathbb{E}\left[\chi^2\left(P_{XY}^{(0)}, Q_{XY}^{(\alpha_0, \alpha_1)}\right)\right], \tag{3}$$

where the expectation is taken over all i.i.d. samples generated from the source and target distributions. Moreover, we define the optimal coefficients

$$(\alpha_0^*, \alpha_1^*) \triangleq \underset{(\alpha_0, \alpha_1) \in \mathcal{A}_1}{\arg\min} \; L_{\text{test}}^{(\alpha_0, \alpha_1)} \tag{4}$$

as our transferability measure, which effectively quantifies the contributions of the source and target tasks in obtaining the optimal performance.

Then, we have the following characterization.

**Theorem 2.** *The testing loss as defined in* (3) *is*

$$L_{\text{test}}^{(\alpha_0, \alpha_1)} = \alpha_1^2 \chi^2\left(P_{XY}^{(0)}, P_{XY}^{(1)}\right) + \frac{\alpha_0^2}{n_0} V^{(0)} + \frac{\alpha_1^2}{n_1} V^{(1)}, \tag{5}$$

---

[3]The assumption on positive entries is without loss of generality, since in practice such joint distributions are typically modeled by some positive parameterized families, e.g., the softmax function.

[4]We shall emphasize that such combination forms are naturally led by the transfer learning model proposed by [4, Section 1] from optimizing a convex combination of Log-Loss, where we refer to Section A of supplementary material for a detailed discussion.

[5]Note that when some $(x, y)$ pair is missing in training samples, we have $Q_{XY}^{(\alpha_0, \alpha_1)}(x, y) = 0$ while $P_{XY}^{(0)}(x, y) > 0$, which would lead to an infinite logarithm risk.

*and the transferability measures as defined in* (4) *are*

$$\alpha_1^* = \frac{\frac{1}{n_0} V^{(0)}}{\chi^2(P_{XY}^{(0)}, P_{XY}^{(1)}) + \frac{1}{n_0} V^{(0)} + \frac{1}{n_1} V^{(1)}}, \quad and \quad \alpha_0^* = 1 - \alpha_1^*, \tag{6}$$

*where, for each* $i = 0, 1$, $V^{(i)}$ *is defined as*

$$V^{(i)} \triangleq \sum_{x \in \mathcal{X}, y \in \mathcal{Y}} \frac{P_{XY}^{(i)}(x, y) \left(1 - P_{XY}^{(i)}(x, y)\right)}{P_{XY}^{(0)}(x, y)}. \tag{7}$$

From (6), and the fact that $V^{(0)} = |\mathcal{X}||\mathcal{Y}| - 1$, the transferability is determined by three key factors: (i) the similarity between source and target tasks, measured by the $\chi^2$-divergence $\chi^2(P_{XY}^{(0)}, P_{XY}^{(1)})$; (ii) the sample sizes $n_0$ and $n_1$ for source and target tasks; and (iii) the model complexity, characterized by the number of model parameter $(|\mathcal{X}||\mathcal{Y}| - 1)$ in $V^{(0)}$.[6]

Current transfer learning algorithm designs often focus on the similarities between source and target tasks, while the sample sizes and model complexity are often not sufficiently addressed. In Theorem 2, we show that the transferability is in fact proportional to the number of model parameters, and is inversely proportional to the number of samples in source tasks and the similarity between source and target tasks. Therefore, for a source task with a complex model or few training samples, even though it is similar to the target task, the knowledge transferable from this source task can still be very limited. Such insight was not well captured in many existing transfer learning algorithms, and our result essentially provides the optimal characterization of the task transferability adjusted by the sample complexity in transfer learning.

The established transferability measure is also related to the optimal bias-variance trade-off [9] of this transfer learning problem. Indeed, note that the bias-variance trade-off in testing loss (5) is tuned by $\alpha_0$ and $\alpha_1$, as

$$L_{\text{test}}^{(\alpha_0, \alpha_1)} = \underbrace{\alpha_1^2 \chi^2 \left(P_{XY}^{(0)}, P_{XY}^{(1)}\right)}_{\text{bias term}} + \underbrace{\frac{\alpha_0^2}{n_0} V^{(0)} + \frac{\alpha_1^2}{n_1} V^{(1)}}_{\text{variance term(s)}}, \tag{8}$$

where the bias term does not decay with the sample sizes $n_0, n_1$, while the variance terms vanish with sufficient samples. Then, the transferability measure corresponds to the coefficients $\alpha_0^*, \alpha_1^*$ that achieve the optimal bias-variance trade-off, such that the testing loss is minimized.

## 2.2 Multi-source Transfer Learning

Theorem 2 can be readily generalized to multi-source transfer learning problems. Specifically, suppose that there are $k$ source tasks, referred to as task $i$, for $i = 1, \dots, k$, and a target task, referred to as task 0. Similarly, for each task $i = 0, \dots, k$, we use $P_{XY}^{(i)}$, $\{(x_\ell^{(i)}, y_\ell^{(i)})\}_{\ell=1}^{n_i}$, and $\hat{P}_{XY}^{(i)}$ to denote the underlying distribution, $n_i$ i.i.d. samples generated from $P_{XY}^{(i)}$, and the corresponding empirical distribution as defined in (1), respectively.

Similar to (2), we consider the convex combination of the models learned from different tasks

$$Q_{XY}^{(\boldsymbol{\alpha})} \triangleq \sum_{i=0}^{k} \alpha_i \hat{P}_{XY}^{(i)}, \quad \boldsymbol{\alpha} \in \mathcal{A}_k. \tag{9}$$

Then, we define the testing loss $L_{\text{test}}^{(\boldsymbol{\alpha})}$ and the corresponding transferability measure $\boldsymbol{\alpha}^*$, as [cf. (3), (4)]:

$$L_{\text{test}}^{(\boldsymbol{\alpha})} \triangleq \mathbb{E}\left[\chi^2\left(P_{XY}^{(0)}, Q_{XY}^{(\boldsymbol{\alpha})}\right)\right] \quad \text{and} \quad \boldsymbol{\alpha}^* \triangleq \underset{\boldsymbol{\alpha} \in \mathcal{A}_k}{\arg\min} \; L_{\text{test}}^{(\boldsymbol{\alpha})}. \tag{10}$$

Similar to Theorem 2, we have the following result for multi-source transfer learning.

---

[6]When evaluating $V^{(1)}$, this quantity is related to the source distribution. From this perspective, the model complexity reflects how hard the task is.

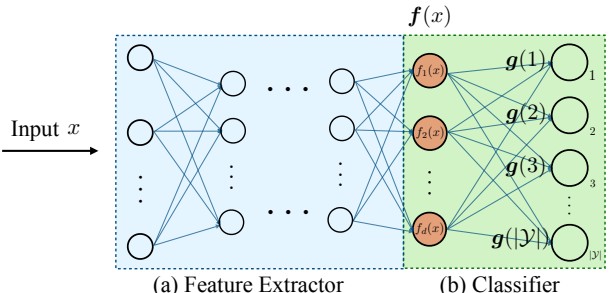



(a) Feature Extractor        (b) Classifier



Figure 1: A pre-trained neural network for classification can be divided into (a) a feature extractor which generates feature $\boldsymbol{f}(x) = [f_1(x), \cdots, f_d(x)]^{\mathrm{T}} \in \mathbb{R}^d$, and (b) a classifier with the weights $\boldsymbol{g}$. With $\boldsymbol{f}$ fixed, our framework optimizes the weights $\boldsymbol{g}$ in the topmost layer for each task, to obtain the corresponding parameterized representation.

**Theorem 3.** *For the model* (9)*, the testing loss under the target task is*

$$L_{\text{test}}^{(\boldsymbol{\alpha})} = \chi^2 \left( P_{XY}^{(0)}, \sum_{i=0}^{k} \alpha_i P_{XY}^{(i)} \right) + \sum_{i=0}^{k} \frac{\alpha_i^2}{n_i} V^{(i)}, \tag{11}$$

*where $V^{(i)}$'s are as defined in* (7)*, for all $i$.*

From Theorem 3, the transferability measure $\boldsymbol{\alpha}^*$ as defined in (10) can be computed by solving a non-negative quadratic programming problem [10]. Similar to the discussions in Section 2.1, such transferability measure quantifies the knowledge transferable from different source tasks to the target task with the sample complexity being considered.

## 3 Parametric Models and Transfer Learning Algorithm

### 3.1 Transferability Measure with Pre-trained Neural Network

This section extends the analyses in the discrete data domain to continuous data in practical problems. In such cases, the previously adopted learning model (1) has infinite parameters due to the infinite cardinality $|\mathcal{X}|$, and thus can not be effectively represented. In order to apply the previous analyzing framework, we first propose a parameterized representation for modeling features of the continuous data by exploiting a pre-trained model.

As shown in Figure 1, a pre-trained network can be divided into two parts: (a) the previous layers for extracting $d$-dimensional features $\boldsymbol{f}(x) = [f_1(x), \cdots, f_d(x)]^{\mathrm{T}}$ from the data variable $x$, and (b) the topmost layer for linear classification, with weights $\boldsymbol{g}(y) = [g_1(y), \cdots, g_d(y)]^{\mathrm{T}}$ indexed by label $y$. When the feature $\boldsymbol{f}(x)$ is given and fixed, the models learned from different tasks can be effectively represented by a finite collection of parameters, i.e., $\boldsymbol{g}(1), \ldots, \boldsymbol{g}(|\mathcal{Y}|)$.

In particular, our framework considers the discriminative model in the factorization form

$$\tilde{P}_{Y|X}^{(\boldsymbol{f}, \boldsymbol{g})}(y|x) \triangleq P_Y^{(0)}(y) \left( 1 + \boldsymbol{f}^{\mathrm{T}}(x) \boldsymbol{g}(y) \right), \tag{12}$$

which is similar to the ones introduced in factorization machines [11] and natural language processing applications [12]. Then, for each task $i = 0, \ldots, k$, we learn corresponding weights $\hat{\boldsymbol{g}}_i$, such that the learned model $\tilde{P}_{Y|X}^{(\boldsymbol{f}, \hat{\boldsymbol{g}}_i)}$ fits the training samples[7]. The weight $\hat{\boldsymbol{g}}_i$ can be formally defined as

$$\hat{\boldsymbol{g}}_i \triangleq \arg\min_{\boldsymbol{g}} \chi^2_{R_{XY}} \left( \hat{P}_{XY}^{(i)}, P_X^{(0)} \tilde{P}_{Y|X}^{(\boldsymbol{f}, \boldsymbol{g})} \right), \tag{13}$$

---

[7]The approach of retraining (fine-tuning) the topmost layer is sometimes referred to as the *retrain-head* method [13], which has also been widely adopted in transfer learning applications.

where the fitness is measured as the referenced $\chi^2$-distance [cf. Definition 1] between the empirical distribution $\hat{P}_{XY}^{(i)}$ and the joint distribution[8] $P_X^{(0)} \tilde{P}_{Y|X}^{(\boldsymbol{f},\boldsymbol{g})}$. For convenience, we adopt a unified reference $R_{XY} \triangleq P_X^{(0)} P_Y^{(0)}$ in fitting different tasks.

From (13), $P_X^{(0)} \tilde{P}_{Y|X}^{(\boldsymbol{f},\hat{\boldsymbol{g}}_i)}$ plays the role in the continuous case corresponding to $\hat{P}_{XY}^{(i)}$ in the discrete case. This allows us to apply previous analyses and focus on the discriminative model $\tilde{P}_{Y|X}^{(\boldsymbol{f},\hat{\boldsymbol{g}}_i)}$'s. Analogous to (9), we consider the convex combination of these discriminative models

$$Q_{Y|X}^{(\boldsymbol{\alpha})} \triangleq \sum_{i=0}^{k} \alpha_i \tilde{P}_{Y|X}^{(\boldsymbol{f},\hat{\boldsymbol{g}}_i)} = \tilde{P}_{Y|X}^{(\boldsymbol{f},\hat{\boldsymbol{g}})} \tag{14}$$

with $\hat{\boldsymbol{g}} \triangleq \sum_{i=0}^{k} \alpha_i \hat{\boldsymbol{g}}_i$. Then, we define the testing loss and corresponding transferability measure as [cf. (10)]

$$L_{\text{test}}^{(\boldsymbol{\alpha})} \triangleq \mathbb{E}\left[\chi_{R_{XY}}^2 \left(P_{XY}^{(0)}, P_X^{(0)} Q_{Y|X}^{(\boldsymbol{\alpha})}\right)\right] \quad \text{and} \quad \boldsymbol{\alpha}^* \triangleq \underset{\boldsymbol{\alpha} \in \mathcal{A}_k}{\arg\min}\ L_{\text{test}}^{(\boldsymbol{\alpha})}, \tag{15}$$

for which we have the following characterization.

**Theorem 4.** *The testing loss* (15) *associated with the model* (14) *is*

$$L_{\text{test}}^{(\boldsymbol{\alpha})} = \chi_{R_{XY}}^2 \left(P_X^{(0)} \tilde{P}_{Y|X}^{(\boldsymbol{f},\boldsymbol{g}_0)}, \sum_{i=0}^{k} \alpha_i P_X^{(0)} \tilde{P}_{Y|X}^{(\boldsymbol{f},\boldsymbol{g}_i)}\right) + \sum_{i=0}^{k} \frac{\alpha_i^2}{n_i} \tilde{V}^{(i)} + \chi_{R_{XY}}^2 \left(P_{XY}^{(0)}, P_X^{(0)} \tilde{P}_{Y|X}^{(\boldsymbol{f},\boldsymbol{g}_0)}\right), \tag{16}$$

*where* $\boldsymbol{g}_i \triangleq \arg\min_{\boldsymbol{g}} \chi_{R_{XY}}^2(P_{XY}^{(i)}, P_X^{(0)} \tilde{P}_{Y|X}^{(\boldsymbol{f},\boldsymbol{g})})$, *and where* $\tilde{V}^{(i)}$ *is a constant independent of* $\boldsymbol{\alpha}$ *characterized in the supplementary material [cf. (32)].*

Moreover, note that from the definition of $\boldsymbol{g}_i$, the joint distribution $P_X^{(0)} \tilde{P}_{Y|X}^{(\boldsymbol{f},\boldsymbol{g}_i)}$ can be interpreted as a projection of $P_{XY}^{(i)}$ onto the distribution family $\left\{P_X^{(0)} \tilde{P}_{Y|X}^{(\boldsymbol{f},\boldsymbol{g})} : \boldsymbol{g} : \mathcal{Y} \to \mathbb{R}^d\right\}$, with referenced $\chi^2$-distance used as the distance measure. Therefore, the terms of (16) share similar interpretations as their counterparts in Theorem 2, with the distances measured in the projected space. Again, $\boldsymbol{\alpha}^*$ can be efficiently computed by solving a non-negative quadratic programming problem.

### 3.2 Multi-source Transfer Learning Algorithm

With our theoretic analyses in Theorem 4, we develop a knowledge transfer algorithm for multi-source transfer learning. Different from the previous analyses where $\boldsymbol{f}$ is fixed, our algorithm jointly optimizes the extracted feature $\boldsymbol{f}$, the weights $\boldsymbol{g}$, together with the combining coefficients $\boldsymbol{\alpha}$ to obtain better performance.

To begin, for given $\boldsymbol{f}$, $\boldsymbol{g}$, and $\boldsymbol{\alpha}$, we introduce the loss function

$$L^{(\boldsymbol{\alpha},\boldsymbol{f},\boldsymbol{g})} \triangleq \sum_{i=0}^{k} \alpha_i \chi_{R_{XY}}^2 \left(\hat{P}_{XY}^{(i)}, P_X^{(0)} \tilde{P}_{Y|X}^{(\boldsymbol{f},\boldsymbol{g})}\right). \tag{17}$$

The following result illustrates that, the $\hat{\boldsymbol{g}}$ can be computed via directly minimizing this loss, without evaluating each $\hat{\boldsymbol{g}}_i$ individually.

**Proposition 5.** *The* $\hat{\boldsymbol{g}}$ *as defined in* (14) *satisfies*

$$\hat{\boldsymbol{g}} = \underset{\boldsymbol{g}'}{\arg\min}\ L^{(\boldsymbol{\alpha},\boldsymbol{f},\boldsymbol{g}')}.$$

---

[8]The joint distribution $P_X^{(0)} \tilde{P}_{Y|X}^{(\boldsymbol{f},\boldsymbol{g})}$ is defined as $\left[P_X^{(0)} \tilde{P}_{Y|X}^{(\boldsymbol{f},\boldsymbol{g})}\right](x,y) \triangleq P_X^{(0)}(x) \tilde{P}_{Y|X}^{(\boldsymbol{f},\boldsymbol{g})}(y|x)$, for all $(x,y)$. Note that when the discriminative model $\tilde{P}_{Y|X}^{(\boldsymbol{f},\boldsymbol{g})}$ is fixed, $P_X^{(0)} \tilde{P}_{Y|X}^{(\boldsymbol{f},\boldsymbol{g})}$ corresponds to the optimal approximation of the target distribution $P_{XY}^{(0)}$.

Table 1: Test accuracies (%) on the target task, with the network trained on samples from single source. All reported accuracies are averaged over 5 repeated experiments.

| Source Task | 1 | 2 | 3 | 4 |
|---|---|---|---|---|
| Acc. on the target task | 66.5 | 59.7 | 56.2 | 77.1 |

Table 2: Test accuracies (%) on the target task, compared with the combining coefficients $\boldsymbol{\alpha}$ determined by 20 rounds of random searches (RS).

| Target Sample Size | 6 | 20 | 100 |
|---|---|---|---|
| Acc. with only target samples | 70.9 | 74.4 | 81.5 |
| Average acc. by 20 RS | 67.8 | 73.9 | 75.4 |
| Highest acc. by 20 RS | 74.4 | 78.0 | 80.8 |
| Acc. by Algorithm 1 | **78.9** | **81.2** | **83.7** |

Then, with training samples from different tasks, our algorithm alternates between two different kinds of optimizations: (i) the optimization of $\boldsymbol{\alpha}$ for given $(\boldsymbol{f}, \boldsymbol{g})$ to minimize the testing loss $L_{\text{test}}^{(\boldsymbol{\alpha})}$ as defined in (16), via solving a non-negative quadratic programming problem; and (ii) the optimization of $(\boldsymbol{f}, \boldsymbol{g})$ for given $\boldsymbol{\alpha}$ to minimize the loss $L^{(\boldsymbol{\alpha}, \boldsymbol{f}, \boldsymbol{g})}$ as defined in (17) via training the neural network. We summarize the procedures as Algorithm 1.

Specifically, it can be shown that both the testing loss $L_{\text{test}}^{(\boldsymbol{\alpha})}$ and the loss $L^{(\boldsymbol{\alpha}, \boldsymbol{f}, \boldsymbol{g})}$ can be represented by some expectations of features $\boldsymbol{f}$ and $\boldsymbol{g}$. In computing these losses, these expectations are approximated by corresponding empirical means, with details provided in the supplementary material.

---

**Algorithm 1** Multi-Source Knowledge Transfer Algorithm

---

1: **Input:** target and source data samples $\{(x_l^{(i)}, y_l^{(i)})\}_{l=1}^{n_i}$ $(i = 0, \cdots, k)$
2: Randomly initialize $\boldsymbol{\alpha}^*$
3: **repeat**
4:    $(\boldsymbol{f}^*, \boldsymbol{g}^*) \leftarrow \arg\min_{\boldsymbol{f}, \boldsymbol{g}} L^{(\boldsymbol{\alpha}^*, \boldsymbol{f}, \boldsymbol{g})}$
5:    $\boldsymbol{\alpha}^* \leftarrow \arg\min_{\boldsymbol{\alpha} \in \mathcal{A}_k} L_{\text{test}}^{(\boldsymbol{\alpha})}$
6: **until** $\boldsymbol{\alpha}^*$ converges
7: $(\boldsymbol{f}^*, \boldsymbol{g}^*) \leftarrow \arg\min_{\boldsymbol{f}, \boldsymbol{g}} L^{(\boldsymbol{\alpha}^*, \boldsymbol{f}, \boldsymbol{g})}$
8: **return** $\boldsymbol{f}^*, \boldsymbol{g}^*$

---

With the $\boldsymbol{f}^*$ and $\boldsymbol{g}^*$ computed by the algorithm, for a newly observed target sample $x$, the predicted label $\hat{y}$ is given by the MAP (maximum a posterior) decision rule

$$\hat{y}(x) = \arg\max_{y \in \mathcal{Y}} \tilde{P}_{Y|X}^{(\boldsymbol{f}^*, \boldsymbol{g}^*)}(y|x) = \arg\max_{y \in \mathcal{Y}} P_Y^{(0)}(y) \left(1 + \boldsymbol{f}^{*\mathrm{T}}(x)\boldsymbol{g}^*(y)\right). \tag{18}$$

## 4 Experiments

To validate the effectiveness of our algorithms in multi-source learning and few-shot transfer learning scenarios, we conduct a series of experiments on common datasets for image recognition, including *CIFAR-10* [14], *Office-31* and *Office-Caltech* [15]. In all experiments, the $\boldsymbol{g}$ in the classifier is simply generated by an embedding layer.

### 4.1 Multi-source Transfer Learning

We conduct multi-source transfer learning experiments on *CIFAR-10*, which contains $50\,000$ training images and $10\,000$ testing images in 10 classes. To begin, we construct the source tasks and target task by dividing the original CIFAR-10 dataset into five disjoint subdatasets, each containing two classes of the original data, which corresponds to a binary classification task. Then, we choose one as our target task (task 0), and use the other four as source tasks for transferring knowledge, referred to as task 1, 2, 3, 4.

Moreover, for each source task, 2000 images are used for training, with 1000 images per binary class, and we set target sample size $n_0$ to $n_0 = 6, 20, 100$, respectively. Throughout this experiment,

Table 3: Test accuracies for target tasks under different transfer settings (source → target) on *Office-31*

| Method | A→D | A→W | D→W | D→A | W→A | W→D |
|---|---|---|---|---|---|---|
| SDT [20] | 86.1 | 82.7 | 95.7 | 66.2 | 65.0 | **97.6** |
| DAMA [21] | 86.3 | 84.5 | 95.5 | 66.5 | 65.7 | 97.5 |
| FADA [22] | 88.2 | 88.1 | 96.4 | 68.1 | 71.1 | 97.5 |
| UDDA [23] | 89.0 | **88.2** | 96.4 | 71.8 | **72.1** | **97.6** |
| Ours | **90.0** | 87.3 | **96.5** | **72.4** | **72.1** | 97.2 |

Table 4: Test accuracies for target tasks under different transfer settings on *Office-Caltech*

| Method | A→C | W→C | D→C | C→A | C→W | C→D |
|---|---|---|---|---|---|---|
| GFK [15] | 68.4 | 68.4 | 64.5 | 83.8 | 78.7 | 74.6 |
| TLDA [24] | 76.1 | 71.0 | 65.4 | 84.2 | 85.2 | 78.9 |
| DTML [25] | 72.0 | 71.6 | 67.1 | 86.0 | 85.0 | 79.6 |
| CPNN [26] | 78.5 | **73.5** | 68.0 | 86.3 | **86.2** | 80.1 |
| Ours | **80.3** | 72.9 | **72.2** | **88.4** | 85.9 | **83.5** |

the feature $f$ is of dimensionality $d = 10$, generated by GoogLeNet [16], followed by two fully connected layers for further dimension reduction.

Unlike common transfer learning settings where the labels for the source and target tasks are closely related, here the binary labels for these 5 sub-datasets are in general irrelevant. Therefore, we first establish the correspondences between labels as follows. For each given source task, we first train the network on its training samples, while the test accuracy is evaluated on the test samples from the target task. Then, we flip the original binary label for this source, if the test accuracy is less than 50%. The resulting test accuracies on the target set are summarized in Table 1.

In our implementation of Algorithm 1, we use the *CVXPY* [17, 18] package for solving the non-negative quadratic programming in line 5. In addition, the alternating iteration is stopped when the element-wise differences for $\alpha^*$ computed in two successive iterations are at most 0.05.

Then, the test accuracies of our algorithm on the target set are shown in Table 2, where we have compared our performance with random search (RS) strategy. Specifically, in the RS strategy, we generate the coefficients $\alpha$ from the log-uniform distribution [19] in $[0.001, 1]$, for 20 rounds. The results indicate that our approach outperforms the random search method. Also, the difference in Table 2 and Table 1 also shows the performance gain of multi-source transfer learning over single-source.

## 4.2 Few-shot Transfer Learning

To validate the effectiveness of our algorithm for few-shot learning tasks, we conduct experiments on Caltech-31 and Office-Caltech datasets. These datasets provide typical transfer learning tasks with few available training samples, where the influence of sample complexity is shown.

### 4.2.1 Caltech-31

Caltech-31 dataset contains images of 31 categories, which come from 3 sub-datasets: **A**mazon (2817 images), **D**slr (498 images), and **W**ebcam (795 images). Then, different transfer settings among these sub-datasets are denoted by the "source → target", as: A→D, A→W, D→W, D→A, W→A, and W→D. We adopt the few-shot transfer learning setting in [20], illustrated as follows. Specifically, 3 target samples per category are used for training, and the training sample size (per category) for source task is set to 20 or 8, depending on whether the source task is Amazon or not. Moreover, we also adopt five train-test splits introduced in [20].

In our experiment, the feature $f$ is a 64-dimensional vector, extracted by a VGG-16 [27] network pre-trained on the ImageNet, succeeded by two fully connected layers for dimension reduction.

Table 3 summarizes test accuracies for target tasks under different transfer settings, where all reported accuracies are averaged over five train-test splits. The results indicate that our algorithm generally outperforms existing few-shot transfer learning methods.

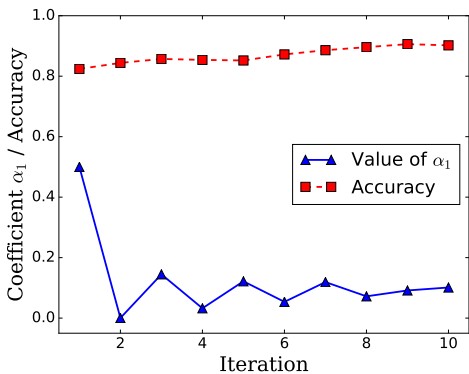

Figure 2: The combining coefficient $\alpha_1$ of the task A→D and test accuracies under testing samples of Dslr during iterations.

In addition, we also investigate the convergence of the coefficient $\boldsymbol{\alpha}^*$ in Algorithm 1. As an example, for the A→D task, the changes of the coefficient $\alpha_1$ (the coefficient of the source loss) and the accuracy on the Dslr test dataset during iterations are shown in Figure 2. From the figure, the value of $\alpha_1$ converges under our stopping criterion, where the optimal testing accuracy is obtained.

### 4.2.2 Office-Caltech

*Office-Caltech* dataset is composed of 10 common categories in *Office-31* and *Caltech-256*, divided as four sub-datasets: **A**mazon (958 images), **C**altech (1123 images), **W**ebcam (295 images), and **D**slr (157 images). We focus on the 6 transfer settings depending on C, i.e., A→C, W→C, D→C, C→A, C→W, and C→D, which have few common categories between source and target tasks.

In addition, we follow the setting introduced in [15] for train-test split. The feature $\boldsymbol{f}$ is of dimensionality $d = 10$, based on the DeCAF feature [28, 29] with 2 fully connected layers for dimension reduction.

Table 4 shows the performance for our algorithm, in comparison with several semi-supervised and few-shot domain adaptation algorithms. It is worth mentioning that, though our approach does not use the unlabeled data samples in training, it provides competitive performance as the semi-supervised algorithm CPNN [26], and can be better on specific tasks.

## 5   Related Work

**Theoretical Analyses of Transfer Learning and Transferability.** Most of the theoretical works about transfer learning focus on deriving upper bounds for the transferability or the performance of transfer learning. For example, the generalization error can be bounded by the VC-dimension of the hypothesis space [30], the total variance distance [4] or the mutual information between training samples and outputs [5], and the Jensen-Shannon distance between domains [6]. The choices of measures are determined mostly by the problem settings. Furthermore, some of these different measures and the $\chi^2$-divergence used in our work are closely related, which are generalized as $f$-divergence [31]. However, there can exist a significant gap between the theoretical bounds and the performance for real tasks [7].

There are also works concentrating on defining a transferability measurement in an empirical way. For instance, the empirical log-likelihood on the target data under the network trained by source samples can measure how much the source samples would help improve the target task [13, 32]. Compared with these works, we establish a transferability measure and provide an analytical expression, for guiding algorithm designs.

**Transfer Learning Algorithms.** Transfer learning algorithms based on the insights from theoretical works intuitively measure the similarities between different domains. The similarity measures include the low-rank common information [33], K-L divergence [34, 35, 36, 37], $l_2$-distance [38, 39, 40], and Wasserstein distance [41]. Additionally, transfer learning problems also share the similar framework with meta-learning, which concentrates on obtaining a generalized model for different

tasks, especially when lacking enough samples for all categories [42]. In comparison with the above works, our algorithm takes the sample sizes and model complexity into consideration, and can be more applicable for general learning tasks, including the few-shot setting.

**Multi-source Domain Adaptation and Few-shot Domain Adaption.** Multi-source domain adaptation considers the approaches of combining multiple tasks together. Conventional methods mainly include instance weighting [43] and domain weighting [44, 15], which re-weight samples or loss functions in training, respectively. Based on deep learning, cutting-edge algorithms attempt to maximize the domain confusion [45, 46] or learn the domain-invariant representations [47].

Semi-supervised and few-shot domain adaptation focus on transfer learning algorithms under few labeled target samples, which is one of our work's application scenarios. Compared with conventional domain adaptation, this field pays attention to embedding samples into an intrinsic low-dimensional subspace [15]. Common algorithms aim at learning domain-invariant representations, including simultaneous deep transfer (SDT) method [20] and semantic alignment method [22, 23].

Compared with these empirical studies, our characterization provides a practical learning algorithm under theoretical guarantees.

# 6   Conclusion

This paper introduces a mathematical framework for quantifying the transferability in multi-source transfer learning problems. Our characterization reveals the essential roles of sample sizes and model complexity in knowledge transferring, which demonstrates potentials in establishing a unified understanding of various transfer learning algorithms. In addition, we develop a multi-source transfer learning algorithm based on the theoretical analyses. Experiments on practical multi-source learning tasks show the effectiveness of our proposed algorithm.

# Acknowledgements

The research of Shao-Lun Huang is supported in part by the National Natural Science Foundation of China under Grant 61807021, and the Shenzhen Science and Technology Program under Grant KQTD20170810150821146.

The research of Lizhong Zheng is supported in part by the National Science Foundation (NSF) under Award CNS-2002908 and the Office of Naval Research (ONR) under grant N00014-19-1-2621.

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
