# OpenReview forum: "A Mathematical Framework for Quantifying Transferability in Multi-source Transfer Learning"
_NeurIPS.cc/2021/Conference — NeurIPS 2021 Poster_

### Official Review · Reviewer_CHLY · 2021-07-14

**Rating:** 7
**Confidence:** 4

**Summary:**

The paper aims to quantify transferability in multi-task in multi-source transfer learning in terms of model complexity, sample size, and similarity of the tasks. The analysis is done for the discrete case and later generalized for a specific model with a fixed embedding structure and a particular readout layer that resembles the factorization machines. The results are validated on a number of transfer learning datasets.

**Limitations And Societal Impact:**

Yes

**Main Review:**

Developing a measure of transferability is an important problem for understanding the dynamics of transfer learning. However, the claims made in the paper are not consistent with the analysis. The paper claims quantifying a notion of transferability in terms of the model complexity and sample size. In statistics and ML, model complexity is generally shown in terms of standard quantities such as Rademacher complexity or VC dimension. Also, sample complexity generally imposes a lower-bound on the number of examples that a certain algorithm requires to learn a particular task. Instead, the provided analysis only shows the optimal loss in terms of the discretized probability densities of the tasks. This quantity is not related to the complexity of the model that is used for transfer learning nor the number of training samples. Therefore, the results do not provide any insights on transferability other than how much the densities of the two tasks are similar.

On the experimental side, it is hard to tell whether the small improvements are significant, given that no error margins are provided.

In summary, the measure of transferability that is developed in the paper is not related to model complexity or sample size. It simply quantifies how close the discretized distributions of two given tasks are to each other. On the experimental side, it is unclear whether this quantity provides any significant improvements.

Minor: the analysis for the binary case is redundant and included in the multi-task setup. This part can be removed without hurting the construction.

=========================================================

After rebuttal:
The authors have addressed most of my questions/concerns regarding the technical details of the paper. The construction in the paper is interesting and non-trivial, especially for the continuous case. Therefore, I would like to raise my score to 7 and vote for acceptance.


**Time Spent Reviewing:**

2

---

> ### Author Response · Authors · 2021-08-10
> **Responses for the comments**
>
> We thank the reviewer for the valuable comments. Our responses are as follows.
>
> First, we would like to mention that there have been a number of model complexity measures developed for different learning objects, e.g., VC dimension, Natarajan dimension,  Pollard's pseudo-dimension, and Rademacher complexity. Specifically, our framework that minimizes the expected testing loss leads to the model complexity, characterized as $(|\mathcal{X}||\mathcal{Y}| - 1)$.
>
> In addition, the sample complexity in general characterizes the dependency of quantities, e.g., the performance of algorithms, on the number of samples. In our development, we are interested in the relationship between the transferability and the sample sizes of different tasks, with detailed characterizations illustrated in, e.g., Theorem 2 and Theorem 4.
>
> Regarding the experimental design, our primary goal is to illustrate how to apply our theoretic transferability measurements to improve the performance of practical learning tasks, which has been demonstrated in the manuscript.
>
> Finally, we would like to mention that, putting the binary case as a separate subsection is for the purpose of demonstration so that the fundamental insights can be illustrated compactly by a simple analytical expression, without the need of introducing complex notations.

---

> > ### Comment · Reviewer_CHLY · 2021-08-11
> > **What does "learned models" mean in line 82?**
> >
> > In Eq (2), you define $Q_{XY}^{(\alpha_0, \alpha_1)}$ which is a convex combination of the empirical distributions $\hat{P}_{XY}^{(i)}$. However, given sets of samples {$x_n, y_n$} for each task, these empirical distributions are simply calculated by summing the frequencies in Eq (1). So my question is, when you say "learned models", what does it refer to? There is nothing learned here (in the sense of model training). These are just empirical distributions and the "model complexity" that you define in Eq (8) basically quantifies how close these two empirical distributions are to each other.
> >
> > Can you please elaborate on this? I might be missing something.

---

> > > ### Author Response · Authors · 2021-08-13
> > > **Responses for the questions**
> > >
> > > We thank the reviewer so much for the questions.
> > >
> > > We would like to clarify that the seemly trivial empirical distributions are the learned models based on the discrete data space. We can use all the entries of the distribution mass function as the parameters of our model. Such a model family contains all possible mass functions in the space and the empirical distribution is the efficient estimator of the underlying distribution. Furthermore, the name 'learned model' is trying to make our theory consistent in Section 2 and 3. In Section 3, the model is parameterized by the feature of data for the continuous space, but these 2 models are both learned from data.
> > >
> > > The statement that 'The "model complexity" that you define in Eq (8) basically quantifies how close these two empirical distributions are to each other.' could be different, since $V^{(0)}$ and $V^{(1)}$ are constants while empirical distributions are random variables. Our model complexity provides such a pattern that when the discrete sample space is larger, it would be more difficult for us to estimate the distribution, which is quite consistent with our intuition and describes how difficult the problem is.

---

> > > > ### Comment · Reviewer_CHLY · 2021-08-17
> > > > **Follow up**
> > > >
> > > > Thank you for your response. Your construction is more clear now. However, I would recommend using a different terminology for the "model complexity": in the discrete case, the model parameters are $\alpha$'s, and what shows up in the bound should be called "task complexity" or something similar. The current terminology irritates readers who are used to definition of these terms in different contexts.
> > > >
> > > > I have one last question before revising my score:
> > > > In Eq. (12), do you assume that you have access to the marginal distribution of $y$, i.e. $P_y^{(0)}$? If yes, how? If not, how does the analysis change?
> > > >
> > > > Minor: line 182, "defined in (13)" (not (14))

---

> > > > > ### Author Response · Authors · 2021-08-19
> > > > > **Responses for follow up**
> > > > >
> > > > > We thank the reviewer so much for the suggestions of the terminology and literature, and we will make improvements in the future version.
> > > > >
> > > > > Regarding the estimation of $P_Y^{(0)}$, we basically in our experiments use its empirical distribution $\hat {P}_Y^{(0)}$. The reason lies in that the empirical distribution is the efficient estimator of $P_Y^{(0)}$. Additionally, $\mathcal{Y}$ is always a discrete space with a rather small cardinality (the number of categories), which means such an estimation can be quite accurate based on i.i.d. assumptions. However, in terms of some extreme cases, there are some other ways of estimating $P_Y^{(0)}$, e.g., the Laplace smoothing method.

---

### Official Review · Reviewer_9X81 · 2021-07-16

**Rating:** 5
**Confidence:** 3

**Summary:**

In this paper, the authors propose a mathematical framework for quantifying the transferability in multi-source transfer learning problem. Specifically, the target model is considered to be a linear combination of the source model, and the combination coefficients is used to denote the transferability. Analyses further show that the transferability is characterized by the sample size, model complexity, and source-target similarity. A parameterized model is then proposed to achieve the quantifiable transferability measure. Experiments on image classification task are done to verify the effectiveness of the proposed algorithms.

**Limitations And Societal Impact:**

An explicity limitation section is encrouged.

**Main Review:**

Here are the comments:

Pros:

(1)	A mathematical framework that establishes a transferability measure quantified by sample size, model complexity, and source-target X^2-distance.

(2)	A parameterized model based on the theoretical framework, and an iterative algorithm for training deep neural network.

(3)	The experiments show the effectiveness of the proposed algorithm in some extent.

Cons:

(1)	The analyses are limited to a stacking-based target model and the proposed X^2-distance, which constrains its generality. The problem setting studied requires the target labelled data. It is more interesting to have discussions on the popular unsupervised domain adaptation setting where the target labels are not available. Moreover, [ref1] studies the similar problem setting (multi-source setting with limited target labelled data) and also proposes a stacking-based target model with the base model coefficients as the domain similarity. It is necessary to discuss with this work.

[ref1] Source-target similarity modelings for multi-source transfer gaussian process regression. ICML. 2017.

(2)	The experiments are done on the classification tasks. Is the analysis applicable to the transfer regression problem?

(3)	Regarding Eq.(12), is it only for the target task (as the P_Y^{(0)}(y) is used)? What is the dimensionality of \mathbf{g}_i? How to obtain P_{XY}^{(0)} and P_{X}^{(0)} in Eq. (15)?

(4)	The improvements of the proposed method compared with the other baselines are not significant as shown in Table 3 and Table 4. Note that some baselines are unsupervised domain adaptation method, which cannot leverage the target labels.

(5)	The empirical evaluation will be more convincing if promising results are obtained on the more challenging dataset, e.g., office-home dataset.


**Time Spent Reviewing:**

5

---

> ### Author Response · Authors · 2021-08-10
> **Responses for the comments**
>
> We thank the reviewer for the valuable comments, and our responses are as follows.
>
> (1) First, we would like to mention that due to the well-established relationships between different information-theoretic measures, e.g., $f$-divergences, one can readily express similar results expressed in other possible measures, with slightly different mathematical properties. Therefore, the usage of chi-squared divergence would not constrain the generality of our framework, and we choose it mainly for mathematical convenience. In addition, we appreciate the reviewer's suggestion in considering more general unsupervised settings, which is indeed our ongoing work. Though the extension on the semi-supervised and unsupervised setting is attempting, it can be difficult to contain all the discussions in a single paper. Thus, this paper focuses on establishing a mathematical framework in the most representative and classical setting, while retaining the applicability to more general settings.   Moreover, we thank the reviewer for pointing out the related literature [ref1], which characterizes the task similarities using covariance matrices. We will add detailed discussions on the connection between our framework and [ref1] in the final version.
>
> (2) As mentioned above, while the presentation of this paper focuses on the classification tasks, the proposed mathematical framework can be easily extended to other settings, e.g., transfer regression problems.
>
> (3)
>
> $\bullet$ In Eq.(12), we choose the marginal distribution $P_Y^{(0)}$ in our parameterized model since the goal is to learn the target task $P_{XY}^{(0)}$.
>
> $\bullet$ All $g_i$'s are $d$-dimensional vectors.
>
> $\bullet$ As we have illustrated in the supplementary material (Section E.), the evaluation of Eq. (15) requires only the computation of some expectations over $P_{XY}^{(0)}$ and $P_{X}^{(0)}$, rather than the distributions. In our implementation, we simply use the empirical means on the target samples as the estimation of corresponding expectations.
>
> (4) In our experiments, we compare our method with some semi-supervised (not unsupervised) approaches, and have guaranteed that the available label information for the two approaches is the same. Indeed, as semi-supervised approaches can make use of extra information from unlabeled data, the reported results demonstrate the effectiveness of our proposed method.
>
> (5) Regarding the datasets, our choices of Caltech-31 and Caltech-Office were based on the practical multi-source learning tasks with few available training samples for the target task. We appreciate the reviewer's suggestions on datasets and would provide related discussions in the final version of the paper.

---

> > ### Comment · Reviewer_9X81 · 2021-08-12
> > **After Rebuttal**
> >
> > Thanks for the authors' response.
> >
> > Regarding (2), as for regression task, the outputs are continuous, it is more challenging to estimate the $P_{XY}^{(0)}$. I am not convinced that it is as easy as the classification task.
> >
> > Regarding (3), as the target domain has limited labelled data, how to gaurantee a good estimation of $P_{XY}^{(0)}$?

---

> > > ### Author Response · Authors · 2021-08-13
> > > **Responses for After Rebuttal**
> > >
> > > We thank the reviewer so much for the questions.
> > >
> > > First, we would like to clarify that it is not necessary to estimate the underlying distribution $P_{XY}^{(0)}$ in our algorithm. The concern from the reviewer is actually our motivation for Section 3. We propose the model based on $d$-dimensional features just for lowering the complexity. Although the testing error is expressed as the chi-square distance between $P_{XY}^{(0)}$ and our model, it actually requires only the computation of some expectations of $f$ and $g$ over $P_{XY}^{(0)}$ and $P_X^{(0)}$, as illustrates in Section E of the supplementary material.
> > >
> > > Then, lacking labelled data is the inherent difficulty for transfer learning. However, the empirical mean is the sufficient statistic for the expectation, which is one of the most reasonable methods to estimate those expectations of $f$ and $g$.

---

### Official Review · Reviewer_HA1J · 2021-07-16

**Rating:** 6
**Confidence:** 3

**Summary:**

The authors propose a mathematical analysis aiming at quantifying the transferability in Transfer Learning.  More specifically, they consider a transferability measure and they show that such a quantity is characterized  by the number of samples, the complexity or dimensionality of the machine learning model and the similarity (according to the \Chi-squared distance) between source and target tasks (see Thm. 2 and Thm. 3). For given source tasks, the authors learn the target task by a class of learning model which linearly combines the models learned from individual tasks by some designable coefficients. Then, they use the optimal combining coefficients achieving the minimum testing loss on the target task as the transferability measure. By doing this, they manage to illustrate the contribution of each model in learning the target task, and effectively quantify the knowledge transferable among different tasks. They start from discrete data setting and then, they extend and adapt the analysis to the continuous data setting by introducing a related transferability measure that can be computed by solving a non-negative quadratic programming problem. Exploiting this, the authors develop an alternating iterative algorithm for training deep neural networks in standard supervised transfer learning settings. The numerical experiments on real image classifications tasks show the effectiveness of the proposed method in comparison to many existing state-of-the-art algorithms in multi-source and few-shot settings.

**Ethical Concerns:**

no ethical issue in my opinion

**Limitations And Societal Impact:**

yes

**Main Review:**

The authors propose an interesting and novel analysis on a topic that could provide new insights on the design of new Transfer Learning algorithms. In particular, they manage to show the natural finding according to which the transferability of a particular source task is typically proportional to the number of samples and the measure of similarity to the target task and it is inversely proportional to the model complexity.

The paper is quite well written.

In lines 28-33, the authors say that “In theoretical analyses [4, 5, 6], sample sizes and model complexity are often included in deriving upper bounds for the transferability or the performance of transfer learning algorithms. However, it is pointed out that such bounds derived under general learning settings are often relatively loose [7], and hence the algorithms designed by directly applying theoretical results can hardly achieve satisfactory performance in practical applications.” Explain to the reader why such bounds are loose. This point is important for understanding the contribution of the paper.

Explain more the intuition of Definition 1. Which is the relation between the similarity measured the authors consider in such a definition and those usually used in standard multi-task and meta-learning framework, such as the variance among the true risk minimizers of the tasks or the complexity of the shared common representation? Is it possible to extend the analysis to more general types of similarity measures, maybe non-heterogeneous measures such as those considered in [1,2,3] below?

Which is the difference between multi-source transfer learning and online meta-learning (see e.g. [4,5,6,7] below) in which we interpret the source tasks as training tasks? Discuss this point and associated literature.

Add a sketch of the proof of the main statements could help the reader in understanding the main tools used during the proofs.

1.   A structured prediction approach for conditional meta-learning, Wang et al. 2020
2.   Conditional Meta-Learning of Linear Representations, Denevi et al. 2021
3.   The Advantage of Conditional Meta-Learning for Biased Regularization and Fine Tuning, Denevi et al. 2020.
4.   Provable Guarantees for Gradient-Based Meta-Learning, Khodak et al. 2019
5.   Online-Within-Online Meta-Learning, Denevi et al. 2019
6.   Learning-to-Learn Stochastic Gradient Descent with Biased Regularization, Denevi et al. 2019.
7.   Adaptive gradient-based meta-learning methods, Khodak et al. 2019

**Time Spent Reviewing:**

48

---

> ### Author Response · Authors · 2021-08-10
> **Responses for the comments**
>
> \response{We thank the reviewer for the valuable comments and suggestions. The responses are as follows.}
>
> *[Question] However, it is pointed out that such bounds derived under general learning settings are often relatively loose [7], and hence the algorithms designed by directly applying theoretical results can hardly achieve satisfactory performance in practical applications.” Explain to the reader why such bounds are loose. This point is important for understanding the contribution of the paper.*
>
> **[Response]** We thank the reviewer for suggesting more detailed illustrations for lines 28-33. Indeed, the looseness of the theoretical bounds has been demonstrated via numerical simulations in *Information-theoretic analysis for transfer learning*[7]. We will add related details in the final version.
>
> *[Question] Explain more the intuition of Definition 1. Which is the relation between the similarity measured the authors consider in such a definition and those usually used in standard multi-task and meta-learning framework, such as the variance among the true risk minimizers of the tasks or the complexity of the shared common representation? Is it possible to extend the analysis to more general types of similarity measures, maybe non-heterogeneous measures such as those considered in [1,2,3] below?*
>
> **[Response]** Regarding Definition 1, the usage of information-theoretic measures (i.e., the chi-squared divergence in Definition 1) allows us to build connections between our framework and classical theoretic characterizations on, e.g., accuracy, or interpretability. To be more specific, previous works have shown that such information measures have deep connections with the objectives in machine learning problems, e.g., the variance of risk minimizers suggested by the reviewer. Due to these connections, our analyzes can be easily extended to machine learning problems with different objectives.
>
> *[Question] Which is the difference between multi-source transfer learning and online meta-learning (see e.g. [4,5,6,7] below) in which we interpret the source tasks as training tasks? Discuss this point and associated literature.*
>
> **[Response]** We also appreciate the reviewer's comments on online meta-learning, which is closely related to our framework. As suggested by the reviewer, the training tasks in online meta-learning serve the role of source tasks in our framework, and our approach can thus be generalized to investigate the performance for online meta-learning, characterized as the sample complexity, model complexity, and the similarities between different tasks. We will add a short discussion in the final version for the sake of completeness.
>
> Finally, we thank the reviewer for the suggestions in improving the presentation of the paper.

---

> > ### Comment · Reviewer_HA1J · 2021-08-11
> > **After Rebuttal**
> >
> > Thanks to the authors for the reply. Could you provide a more specific answer to my question below, please? Your answer is quite vague. I think this is an important point to discuss in order to relate the proposed work to the state-of-the-art literature.
> >
> > Explain more the intuition of Definition 1. Which is the relation between the similarity measured the authors consider in such a definition and those usually used in standard multi-task and meta-learning framework, such as the variance among the true risk minimizers of the tasks or the complexity of the shared common representation? Is it possible to extend the analysis to more general types of similarity measures, maybe non-heterogeneous measures such as those considered in [1,2,3] below?

---

> > > ### Author Response · Authors · 2021-08-13
> > > **Responses for After Rebuttal**
> > >
> > > We thank the reviewer for the detailed explanation of the concerns and would like to share our thoughts as follows.
> > >
> > > First, regarding the similarity measures, the choices of measures are determined mostly by the problem itself, and these different similarity measures are indeed closely related. For instance, the chi-square divergence used in our framework can be bounded by KL divergence, total variation, etc, with comprehensive discussions provided in, e.g., *P. Harremoms and I. Vajda. On pairs of f-divergences and their joint range. IEEE Trans. Inf. Theory, 57(6):3230{3235, Jun. 2011.* Applying similar approaches, we can also establish the connections between our similarity measure and the ones mentioned by the reviewer, e.g., the variance of the true risk minimizers of the tasks, or the complexity of the shared common representation.
> > >
> > > Additionally, our paper is highly related to [ref1], following the empirical risk minimization (ERM) principle and a linear representation, but is different in analyzing the excess risk (testing loss in our paper). Specifically, the difference lies in whether to analyze the excess risk depending on the loss function of training loss. To provide general analyses for any possible $l$ in [ref1], an upper bound based on other distance measures (kernel in [ref1] that might be quite different from $l$) is provided. However, the bound based on local analyses for analyzing the sample complexity $O(n^{1/4})$ might be inaccurate for different kernels and loss functions. As illustrated in the introduction of our paper, our main improvement lies in avoiding analyses depending on bounds. Similarly, [ref2] and [ref3] use the performance bounds based on the assumptions of convexity and l-Lipshitz, which are much more classical assumptions in machine learning theory, but do not avoid the looseness of bounds. These upper bounds in [ref1-3] provide chances for general analyses for evaluating the performance (do not need to define a specific loss or kernel), but sacrifice the precision.
> > >
> > > As for the last question of whether our framework can be extended to other similarity measures, we have a positive answer and it is indeed our consideration in the next step.

---

### Official Review · Reviewer_thNZ · 2021-07-16

**Rating:** 6
**Confidence:** 3

**Summary:**

The paper introduces a transferability measure that takes into account the task similarity as well as the sample size and model complexity. The theoretical formulation is then applied to develop an iterative algorithm to train deep neural networks in a supervised transfer learning setting. It is further shown that the algorithm can be applied to multi-source transfer.


**Limitations And Societal Impact:**

yes

**Main Review:**

**Originality**
The proposed mathematical formulation is novel as it takes into account the sample size and model complexity on top of task similarity which is lacking in previous methods. However, the relation between recently introduced transferability measures e.g. LEEP (Nguyen et al. ICML 2020), NCE (Tran et al. ICCV 2019), and their derivatives are neither discussed nor cited. The results are compared to domain adaptation methods making it unclear whether this work is more suitable as a domain adaptation paper or transferability.

**Quality**
I appreciate the author's effort in deriving a mathematical formulation of transferability that takes into account sample size and model complexity. However, the experiments performed in the paper do not seem sufficient due to the following reasons:

1. The method introduces a transferability measure. Therefore ideally it should be compared to methods such as LEEP and more recent state-of-the-art transferability measures. In case, the proposed method is not suitable to be compared with these methods, this should be clearly explained in the paper.
2. In the current results, the datasets used to evaluate the performance are more suitable for domain adaptation. A more recent standard benchmark VTAB (Zhai et al. arxiv 2019) might be more suitable or the ones used in LEEP, NCE papers making the results of the current method comparable to state-of-the-art.
3. Even in the domain adaptation benchmark the methods compared are not current state-of-the-art e.g. in Office Caltech dataset Wang et al. AAAI 20 is current state of the art. IT is not necessary to beat state-of-the-art methods but at least these methods should be mentioned and the relation between these methods to the current method should be discussed.

**Clarity**
The mathematical formulation is described well and augmented with the algorithm. Although some parts of the description may be improved e.g. in line 109 it is not clear why that term accounts for model parameters.

The related literature section lacks citations to multiple relevant papers as mentioned above making it difficult to assess the contribution of the current paper when put in context with recent papers on transferability measures.

**Significance**
Theoretical estimates of transferability are crucial in order to assess the application of off-the-shelf models in real-world scenarios. In this aspect, the paper does a good job in obtaining an estimate that takes into account the sample size of source and target tasks which is a crucial factor.

However, the experiments performed in the current version do not provide sufficient evidence of the applicability of the proposed measure.

**Time Spent Reviewing:**

8

---

> ### Author Response · Authors · 2021-08-10
> **Responses for the comments**
>
> We thank the reviewer for the valuable comments.
>
> First, we appreciate the reviewer for pointing out the related literature on the transferability measure. Compared with the transferability measures developed in empirical and heuristic manners, e.g. LEEP (Nguyen et al. ICML 2020), NCE (Tran et al. ICCV 2019), our primary goal is establishing a mathematical framework for quantitative analyses of transfer learning problems. Therefore, our experiments design focuses more on illustrating how to apply our theoretic transferability measurements to improve the performance of practical learning tasks, instead of providing comparisons with a comprehensive list of approaches, such as
>
> *Sinapov, J., Narvekar, S., Leonetti, M., and Stone, P. Learning inter-task transferability in the absence of target task samples. In International Conference on Autonomous Agents and Multiagent Systems, pp. 725–733, 2015.*
>
> *Conditional Meta-Learning of Linear Representations, Denevi et al. 2021*
>
> Another practical concern is that a direct comparison among different transferability measures does not make much sense, since these measures in general have different scales. These considerations motivate us to focus on only design experiments in the present form.
>
> In addition, regarding the datasets, our choices of Caltech-31 and Caltech-Office were based on the practical multi-source learning tasks with few available training samples for the target task. We appreciate the reviewer's suggestions on datasets and would provide related discussions in the final version of the paper.
>
> Finally, on the detailed question regarding line 109, we would like to comment that $|\mathcal{X}||\mathcal{Y}| - 1$ is the number of parameters required to describe a joint distribution on $\mathcal{X} \times \mathcal{Y}$.

---

> > ### Comment · Reviewer_thNZ · 2021-08-24
> > **Response to author's comments**
> >
> > I would like to thank the authors for their responses and clarifications.
> >
> > >First, we appreciate the reviewer for pointing out the related literature on the transferability measure. Compared with the transferability measures developed in empirical and heuristic manners, e.g. LEEP (Nguyen et al. ICML 2020), NCE (Tran et al. ICCV 2019), our primary goal is establishing a mathematical framework for quantitative analyses of transfer learning problems. Therefore, our experiments design focuses more on illustrating how to apply our theoretic transferability measurements to improve the performance of practical learning tasks, instead of providing comparisons with a comprehensive list of approaches
> >
> > Thanks for the clarification. I think the above points need to be discussed in the related works section more clearly and in depth. I would encourage the authors to further clarify the exact differences between the measures and where and why exactly methods like LEEP can not be applied.
> >
> > >Another practical concern is that a direct comparison among different transferability measures does not make much sense, since these measures in general have different scales.
> >
> > I am not sure if I understand this correctly. For example, if one transfers from source tasks A and B to a target task T. Based on LEEP scores we can assign alphas for transfer learning in Table 2 and then compare to alphas obtained by the presented method. Can you please explain this in detail?
> >
> > Overall, I think the paper in its current form still needs more clarity in framing to emphasize the key contributions of this paper. I believe discussing the similarities and differences to relevant literature in transferability measures, meta-learning , and domain adaptation would help. Also, the choice of datasets used needs to explained to clarify why the commonly used datasets in transferability and meta-learning were not used.

---

> > > ### Author Response · Authors · 2021-08-26
> > > **Responses for the comments**
> > >
> > > We would like to thank the reviewer for the suggestions.
> > >
> > > Regarding the comparison with LEEP scores, our transferability alpha is a coefficient between 0 and 1, while LEEP is the average log-likelihood of the classifier. We could indeed normalize the LEEP scores to $[0,1]$ for comparing with $\alpha^\star$. We shall expect they have some similarities in tendency, where the result will depend on the way of normalization. However, there could be multiple ways of normalization and we could not theoretically decide which one makes sense, while the coefficient $\alpha^\star$ is theoretically optimal.
> > >
> > > We will add similar discussions in the future version.

---

> > > > ### Comment · Reviewer_thNZ · 2021-09-01
> > > > **Response to author's comments**
> > > >
> > > > I would like to thank the authors for the clarification. Most of my concerns have been addressed.
> > > >
> > > > However, it is still difficult for me to be convinced whether it would be possible to make the contributions of this work clearer without seeing the revision. Therefore, I would again encourage the authors to put more effort into the revision to clarify similarities and differences to relevant literature in transferability measures, meta-learning , and domain adaptation.
> > > >
> > > > Taking into consideration the authors' responses I have increased my rating to 6.

---

### Decision · Program_Chairs · 2021-09-28

**Decision:**

Accept (Poster)

**Comment:**

In this work, a novel mathematical formulation is proposed for the problem of multi-source transfer learning. The theoretical analysis is interesting. Empirical evaluation is good but can be more convincing by conducting experiments on more challenging datasets.

In the rebuttal, most of the concerns raised by reviewers have been well addressed. Though one reviewer sticks to his originally negative rating of 5, I found that the authors have given clear explanations in their responses. Therefore, I recommend an acceptance for this work.

However, the authors are encouraged to make the descriptions clearer in the revision based on reviewers' comments as well as the authors' responses in the rebuttal.

**Consistency Experiment:**

NeurIPS has a long history of experimentation. In 2014, NeurIPS ran an experiment in which 10% of submissions were reviewed by two independent committees to quantify the randomness in the review process. This year, we repeated a variant of this experiment to see how the quality of the review process has changed over time.  This paper was part of the experiment and was therefore assigned to two committees (consisting of reviewers, an Area Chair, and a Senior Area Chair) that reached independent decisions.  If both committees made the same recommendation, this recommendation was followed. If a single committee recommended acceptance, the paper was accepted (with the exception of a few cases in which the other committee identified what we considered a fatal flaw, e.g., an error in a key result).

This copy’s committee reached the following decision: **Accept (Poster)**

The other committee assigned to the paper recommended **Reject**.  You can find the other set of reviews, along with any follow up discussion with the authors here:
https://openreview.net/forum?id=wQZWg82TWx